# Validation Study of a Value-Based Digital Health Questionnaire

**DOI:** 10.3390/ijerph19127034

**Published:** 2022-06-08

**Authors:** Roberto Nuño-Solinís, Elena Urizar, Marisa Merino, Jaime Del Barrio, María Errea Rodríguez

**Affiliations:** 1Fundación Gaspar Casal, General Díaz Porlier, 78, 8ºA, 28006 Madrid, Spain; roberto.nuno@fgcasal.org; 2Deusto Business School Health, University of Deusto, 48014 Bilbao, Spain; elena.urizar@deusto.es; 3Asociación Salud Digital, Calle de Serrano, 90, 28006 Madrid, Spain; marisa.merinohernandez@osakidetza.eus (M.M.); jaimedelbarrio@salud-digital.es (J.D.B.); 4Independent Researcher, 31014 Pamplona, Spain

**Keywords:** value-based digital health, value-based healthcare, questionnaire validation, healthcare organizations

## Abstract

The paradigm of value-based health care is spreading worldwide; however, Value-Based Digital Health (VBDH) is still an emerging concept. VBDH is understood as the use of digital tools to facilitate the generation of value in health. It is accelerated by technological change, cultural, and organizational factors. An accurate diagnosis of the organizational VBDH maturity is crucial to define and implement strategic actions to progress with VBDH transformation. This study aimed to validate a VBDH questionnaire, which measures the degree of maturity of VBDH from the perspective of managers (N = 146) in Spanish healthcare organizations. Results show good internal consistency of the questionnaire. Factor analysis identified seven dimensions to measure VBHC maturity: (1) Resources, incentives, and financing; (2) Knowledge and participation of patients and workers in the strategy of progress towards VBDH; (3) Training of professionals and tool knowledge for advancement in VBDH; (4) Innovation initiatives; (5) Information and its quality; (6) Leadership, strategy and governance; and (7) Knowledge of the fundamentals and objectives, as well as access to relevant VBDH information. The questionnaire presents good validity and internal consistency and meets the requirements to be an instrument for routine use to assess VBDH organizational maturity.

## 1. Introduction

Value-based Digital Health is a construct that arises from the interconnection between Digital Health [1] and Value-based Health Care [2]. Value-Based Digital Health is understood as the use of digital tools, solutions, strategies, and ecosystems to contribute to the generation of value in health [3].

The generation of digital value is a topical debate in the academic literature. There are multiple areas where the benefits of digitization in healthcare are tangible, such as contributing to more efficient and safe processes, reducing administrative burdens, improving clinical decisions, personalizing care, providing remote diagnostic and therapeutic services, providing the patient with new ways of access and communication, improving outcomes and patient, supporting the management of population health and public health strategies, and even contributing to social welfare and planetary health, and reducing energy consumption and unnecessary travel. There are also dysfunctional effects and risks that need to be reduced in its implementation.

The digital transformation in healthcare delivery organizations is driven by an accelerated technological change. All agents in the society are key in the consolidation of this process: health professionals and citizens have “the last word” in its progress and, in particular, citizens are demanding this change. Likewise, the so-called digital divide has been reduced dramatically in the last decade (which does not preclude taking special care to avoid generating inequalities in this area). In short, patients are evolving, being more digital. Patients are more and more demanding of health service interaction through technology, similar to what they do with other services. New non-face-to-face communication channels need to be normalized, and access to their health data and participation in their treatment and care is also a request. All this should be considered without forgetting the barriers imposed by a framework of data protection; this becomes an inevitable journey of the provision of many health services through digital health.

In this context, another movement is revolutionizing healthcare management. Value-based healthcare (VBHC) is now a global trend in many industrialized countries. The concept of value is defined as outcomes that are important to patients relative to the costs required to achieve those outcomes [4]. VBHC seeks to “fix” health systems [5]. Although this purpose is very ambitious and its achievement uncertain, it can be stated that it incorporates relevant elements that had been neglected until the date in the daily management of health organizations. The VBHC paradigm implies keeping the focus on the systematic measurement of the final results of health interventions (including their costs) and incorporating the patient’s perspective, as well as overcoming the technocentric vision and breaking with the habitual complacency of putting the focus only on the activity, average length of stay, processes, resource allocation indicators, and the like. Further, there is a strong commitment to standardize, systematize, and incorporate this way of operating into the clinical and management routine. However, all this is, today, unapproachable without a solid digital transformation strategy aligned with that purpose.

Within the VBHC proposal, the standardized and routine measurement of results and costs that allow progress towards value-based management has become the central axis on which other proposals of the model pivot. Making progress in this approach requires a degree of organizational development and maturity that is, unfortunately, not available in all environments.

The concept of organizational maturity has a long history in the field of quality improvement since the 1979 seminal works of Crosby [6], and specifically in health there is a wide range of instruments aimed at knowing the preparation of organizations in terms of resources, processes, infrastructures, culture, etc., against various concepts and strategies [7,8]. Measuring organizational maturity allows guiding the progress of improvement strategies, which contributes to improving organizational performance [9]. Despite the long life of this concept, the VBDH construct has not yet been defined or articulated in research, management or policymaking in any country. It is, however, assumed, that digital transformation may facilitate the transition towards VBHC, requiring, among other factors, cultural and behavioral changes in providers, payers, and suppliers, acknowledging the current health system’s pitfalls, and widely embracing digital health to disrupt and shift the value-proposition favoring patients [10]. In VBHC, it is difficult to move beyond concept, even in most advanced health systems in the field [11].

Taking into account the above, a questionnaire was developed to measure organizational maturity around the VBDH concept. Organizational maturity in this questionnaire is assessed, from the perspective of managers and directors in health organizations (the questionnaire’s target population), through a series of statements (questionnaire items) that help to evaluate their VBDH knowledge in the organization in which they have management capacity. The questionnaire items are organized or structured into five blocks that, through the items in each block, represent the dimensions of organizational maturity. These are: (1) Leadership and strategy; (2) Culture and supportive environment; (3) Processes and practices; (4) Learning and assessment systems; and (5) Incentives and financing. The questionnaire was designed by a research team coordinated from Deusto Business School Health. The questionnaire was inspired by extant instruments. Despite the existence of validated questionnaires to measure organizational maturity related to several constructs [7,8], such questionnaires have either covered only specific dimensions, such as the organizational learning capability scale [12], or have been implemented in healthcare organizations that work with specific populations of patients, such as the Instrument for the Evaluation of Care Models for Chronicity (IEMAC questionnaire) [13], which measures and identifies aspects where there is room for improvement in healthcare organizations that work with people with chronic conditions. In addition, although there are tools for the measurement of the level of satisfaction of healthcare professionals with telemedicine [14], as well as organizational maturity or readiness for different aspects within value-based (health) care [15], there is no validated questionnaire for measuring the level of VBDH organizational maturity.

This study aims to show the results of studying the validation of a VBDH questionnaire whose purpose is to measure the degree of organizational maturity related to VBDH The goal of this paper is thus twofold. On the one hand, a VBDH questionnaire is presented. On the other hand, its validity to be used as a routine instrument for collecting data from healthcare organizations and its management personnel, based on factor analysis and measurement of its internal consistency, is analyzed and presented. The main hypothesis is that the questionnaire satisfies the validity and consistency of psychometric properties, and therefore, it is ready to be used as an instrument for routinary data collection to understand the level of VBDH organizational maturity. A secondary hypothesis is that the questionnaire is also valid for measuring other aspects, and that different subscales, always within the context of VBDH, are also valid and can be deployed.

## 2. Materials and Methods

Qualitative methods were used to obtain, refine, and prioritize the elements (items) to be included in the questionnaire. It began with a scoping review of the literature, in which key issues to consider were identified [16]. A second phase was dedicated to the refinement of themes through semi-structured telephone interviews with key informants. This phase was followed by a series of conceptual validation sessions led by a panel of experts from different fields, which included the head of an Integrated Care Organization, the general director of the largest patient association in Spain, a researcher from health services, and an Academic Expert in Organizational Theory. In this phase, the experts prepared a draft of questions that were grouped into five domains, which were identified from the review of articles that combined four theoretical currents: Value-based healthcare, Digital health, Integrated care [17], and the learning organization [18].

To demonstrate the validity of the proposed questionnaire, a series of psychometric properties, such as internal consistency and validity of the questionnaire, were evaluated. Factor analysis was conducted. This technique also provides information that allows understanding the internal structure of the questionnaire.

### 2.1. The VBDH Questionnaire

The questionnaire consists of a set of sociodemographic questions followed by 41 items, in which the participants show their degree of agreement/disagreement (on a 5-level Likert scale) with a series of statements that concern the maturity of their organization regarding VBDH and a few open questions for the respondents to provide their opinion regarding some statements related to mechanisms to advance VBDH. The 41 main items are presented to participants in five blocks: (1) Leadership and Strategy, (2) Culture and Enabling Environment, (3) Processes and Practices, (4) Learning and Assessment Systems, and (5) Incentives and Funding. Each block contains different items organized in dimensions or subscales for the measurement of the level of VBDH organizational maturity. The domains and items were refined and reviewed by the research team to create the final questionnaire. An expert panel checked the user interface and conceptual validity of the tool [16]. Thus, the objective is to demonstrate the validity of the questionnaire to be used as a routine tool for data collection in healthcare organizations. To achieve this, this work aims, first, to demonstrate that all items are comprehensible by non-experts (as experts have already validated it), and, second, to identify other subscales that could emerge with the combination of items from the questionnaire, in order to have more dimensions for the future measurement of other VBDH aspects within healthcare organizations.

### 2.2. Sample and Data Collection

To carry out the validation study and to conduct the factor analysis, a collection target of around 150 responses was set a priori before data collection. A secondary objective in collecting the data was that the sample represented participants from all of the autonomous communities of the country.

For data collection, a Google form (Google forms) was used and distributed to a convenience sample of around 900 healthcare managers from all Spanish regions. Finally, a total of 146 responses were collected. The respondents were approached by the research team and had previously agreed to participate in the study before answering the questionnaire.

### 2.3. Validation Analysis of the VBDH Questionnaire

The validity property aims to measure the ability of an instrument to adequately measure a theoretical construct (in our case the VBDH concept). It represents the degree to which a measurement is consistently related to other measurements, with the theoretical hypotheses that define the phenomenon or construct to be measured. To evaluate the validity of the questionnaire, an exploratory factor analysis is proposed.

The factor analysis aims to provide information regarding the relationship between the items of the questionnaire, based on the responses of the participants. This will show a common meaning between the items based on their degree of correlation. The objective of this analysis is to offer, on the one hand, the optimal number of factors within the questionnaire, and, on the other hand, the list of items that will make up each factor. In other words, the exploratory analysis will offer an alternative structure/organization of the questionnaire or subscales within the questionnaire, which, having demonstrated its validity, could be used for different independent purposes, depending on the organizations’ needs.

## 3. Results

### 3.1. Descriptive Results

More than half of the participants were men (N = 84; 57.53%). Most respondents were 45–64 years old (N = 121, 82.88%), and almost half of the participants rated their knowledge about VBDH as “Intermediate”. Only 22.6% of the participants rated their knowledge about VBDH as “High”, and much lower was the percentage who stated their knowledge was “Very high” or “None”. Detailed results are shown in Table 1 below.

Figure 1 shows the distribution of responses (degree of agreement/disagreement) for each of the items of the proposed questionnaire. A high degree of agreement stands out with items 1 and 9 (more than 70% agreed or totally agreed), 2 (more than 60% agreed or totally agreed), and 30 and 31 (in these last two, about 50% agreed or strongly agreed). There was also a high percentage of participants who say they agreed or totally disagreed with items 38, 39, and 40 (52.74%, 52.05%, and 57.54% respectively).

### 3.2. Cronbach’s Alpha: Internal Consistency of the Entire Questionnaire

Cronbach’s alpha test showed a good degree of interrelation and coherence between the items of the questionnaire. All items had Cronbach’s alpha values greater than 0.887, and the overall value was 0.891, which is indicative of good internal consistency of the questionnaire items individually and as a whole. All items, therefore, were understandable and can be maintained in the questionnaire. The results are shown in Table 2 below.

### 3.3. Exploratory Factor Analysis: Validity Analysis of Subscales within the Questionnaire

The results of the exploratory factor analysis of the structure of the questionnaire conducted are shown below. The aim is to understand if items groups or subscales could be generated (factors) to measure different aspects of VBDH in healthcare organizations. The detailed analysis below shows how to obtain the optimal structure, in terms of the number of dimensions and the best factors to form each of those dimensions.

In a first analysis with all the items of the questionnaire, it was observed that some items would be of little relevance if the questionnaire were to be used for this purpose. The first criterion used for the selection of the less relevant items was that the load of the items associated with the factor that integrates them must be greater than 0.4. The following items were excluded for the reason of the factor load < 0.4: items 1, 7, 11, 13, 14, 15, 16, 20, 22, 23, 24, 32, and 41. The same analysis was conducted with the remaining items, and the number of optimal factors was reduced to 8–10 factors. However, for the new analysis, with a lower number of items, a factor load criterion of 0.3 was required. After looking at the results, items 18, 19, 21, and 25 were excluded. The reasons for the exclusion of these items were the following: 1. they belong to the factor with the lowest overall Cronbach’s alpha value; 2. the average inter-item correlation was less than 0.3, indicating a poor relationship between the items. These were the last group of items excluded, and the final analysis was done with the remaining 24 items.

Figure 2 below, the sedimentation graph, shows the optimization result with the 24 items included. The optimal number of factors in this case was 7–8 factors. This means that the first 7–8 factors had variances (eigenvalues) greater than 1. The eigenvalues changed less markedly when more than 7–8 factors were used. Therefore, 7–8 factors seemed to explain most of the variability in our data. The sedimentation plot shows that the first seven factors explained most of the total variability in the data. The remaining factors represented a very small proportion of the variability and are probably not as important.

In the next step, the analysis shows which of the 24 remaining items belonged to each of the seven factors to be extracted. The final results are shown in Table 3. Factor 1 should be interpreted as the most important factor or subscale within the questionnaire, since on average the items comprising it were those with the greatest factor load. In addition, it would also be a factor with high internal consistency, with a Cronbach’s alpha value greater than 0.7. Likewise, the rest of the factors were ordered, with factor 2 being the second most important, and so on up to factor 7, which would be the least important. All the factors had a Cronbach’s alpha greater than 0.5, which indicates a good internal consistency of the items comprising them.

The factor analysis identified seven dimensions or subscales within the questionnaire. These subscales can be used to measure different aspects of VBDH in health organizations:Resources, incentives, and financing.Knowledge and participation of patients and workers in the strategy of progress towards VBDH.Training of professionals and knowledge tools for advancement in VBDH.Innovation initiatives.Information and quality of information.Leadership, strategy, and governance.Knowledge of the fundamentals and objectives, as well as access to relevant information for the VBDH.

These dimensions could be used to measure the progress of health organizations in VBDH in these aspects.

## 4. Discussion

This work presents the results from a validation analysis of a VBDH questionnaire for healthcare organizations in Spain. The conducted analysis validated a VBDH questionnaire that can now be used as a routine tool for data collection by healthcare organizations. The questionnaire will serve to find out the degree of maturity and progress in VBDH of managers and directors working at healthcare organizations. Although there are instruments to measure organizational maturity in healthcare organizations, this questionnaire is pioneer, being the first tool for measuring the level of VBDH organizational maturity.

The questionnaire validation was also part of a larger study. The study aimed to (1) Explore the literature to have a good understanding of the conceptualization of VBDH, (2) Design and validate a questionnaire for the measurement of VBDH maturity of healthcare organizations, and (3) Have a roadmap for the next five years to advance VBDH. The questionnaire validation is central within the objectives of the project, given that having a questionnaire to collect information regarding the VBDH maturity increases the chances of understanding how to advance VBDH.

Data were collected from a sample of 146 managers in organizations that provide healthcare services. All the autonomous communities of the country were represented. Sufficient information was collected to carry out an exploratory factor analysis that provided the structure of the questionnaire (optimal number and composition of factors) as well as allowing a validation study to be carried out. The Kaiser–Meyer–Olkin test of sampling was adequate (KMO = 0.773). Further, correlations were tested and demonstrated to be high between variables. Authors have suggested different sample sizes that can be considered acceptable for conducting factor analysis. Studies support a sample size of a minimum two subjects per variable [19], five subjects per variable or 100 subjects, whichever is larger [20], 100 subjects if the questionnaire demonstrates a clear structure, although more is always better, 300 subjects, though fewer works if correlations are high among variables [21], and others base it on a ratio of cases to the number of factors concluding that 20 subjects per factor is a sufficient sample size [22]. Given that the questionnaire demonstrated, according to a panel of experts, a clear structure, the aim was to have at least 100 responses.

The degree of interrelation and coherence between the items of the questionnaire was measured by Cronbach’s alpha coefficient. Although interpretation of this coefficient is subjective and depends on what is to be validated, it is usually considered high when the value is 0.70 or higher, acceptable–moderate for values between 0.6 and 0.7 [23], and acceptable for values between 0.45 and 0.60 [24].

The results showed a good internal consistency of the questionnaire, demonstrated by the high interrelation found between its items, measured by Cronbach’s alpha. The exploratory factor analysis conducted also provided an additional structure for the questionnaire, ordered according to the importance of the factors that make up its structure. In this case, the first two factors would be the most important in the questionnaire, and therefore the items that make up these two factors are also important, all of them related to resources and financing/incentives and the knowledge and participation of patients and workers in the strategy of moving towards VBDH. However, the five additional dimensions demonstrated they could also be used to measure other relevant aspects to measure the progress of organizations towards VBDH, given the acceptable level of internal consistency demonstrated in the items that comprise them. The descriptive analysis and specifically the distribution of responses to the 41 questionnaire items also complemented the factor analysis and helped with the combination of items when more than one option for item combinations was possible. The reader can quickly check the consistency (in terms of the similarity in the distributional pattern of responses) between the new subscales that resulted from the factor analysis and the distribution of responses for each of the items in each of the new subscales.

The analysis is not without limitations. On the one hand, it is a convenience sample. Further research should focus on collecting data with the same questionnaire in similar populations to check its reliability, a property that could not be evaluated in this work, as information at two different moments of time in comparable populations would be required. On the other hand, the sample is relatively small, having obtained only 146 responses. Although such a sample size is sufficient for the aim of this work—to validate the internal consistency of the questionnaire and its construct validity through an exploratory factor analysis—this sample is not possible to obtain reliable results on what the opinion of the respondents is in relation to the various aspects of VBDH, as it is not representative of the target population at the national level. However, the factor analysis shows how the factors structure is stable. In a first stage, including all the 41 items, the factor aggregation remained similar to the structure that remained when a more conservative approach—excluding some items according to a low factor load criterion—was selected. Some factors that had only one item were allocated to another factor, but when two items belong to one factor, generally they stay together, even if allocated to a different factor. This demonstrates the stability of the factor structure. Although this could be tested with a representative sample, results suggest that no large changes in the way items aggregate would be expected. However, it is true that with a larger sample, more factors will appear, as less items will have to be removed due to the low factor load, probably the result of the small sample size in this study. Finally, most of the respondents (80%) were aged 45–64 years old, so results cannot be generalized to populations of all ages. However, the target population consisted of managers and directors of healthcare organizations, and thus, this was not a surprise, as this age is the most common among people in such position, especially in hospitals. This coincides with previous literature findings [25,26]. Having validated the questionnaire, this work encourages healthcare organizations to use this questionnaire for the purpose for which it has been validated. Having data collected every year may be a good manner of analyzing changes in the VBDH maturity level of healthcare organizations but also could help understanding in which aspects there is room for maturity improvement. This will be one way to design appropriate strategies to improve the level of organizational maturity of those aspects that show weaker maturity levels.

## 5. Conclusions

In conclusion, we can affirm that the analyzed VBDH questionnaire presented good validity and internal consistency. This questionnaire met the basic requirements to be accepted as an instrument for routine use to collect information on the opinion related to VBDH from the perspective of managers in healthcare delivery. However, reliability of the questionnaire remains to be evaluated. This work provides healthcare organizations a questionnaire for data collection capable of providing information on the VBDH maturity level of healthcare organizations. This questionnaire presents one more way to design appropriate strategies to improve the level of organizational maturity of those VBDH-related aspects that show weaker maturity levels in healthcare organizations. Measuring organizational maturity allows guiding the progress of Value-based Digital Health improvement strategies.

## Figures and Tables

**Figure 1 ijerph-19-07034-f001:**
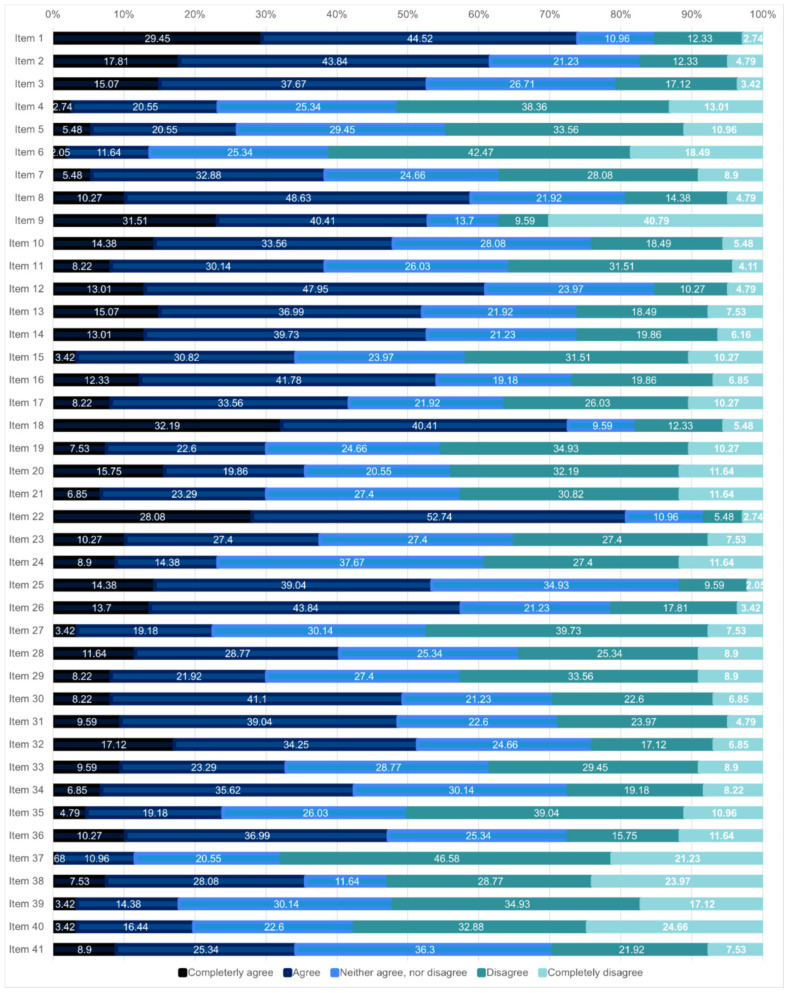
Degree (%) of agreement with the 41 items of the VBDH questionnaire. The item description (translated to English) is provided in Table 2 below, and the original questionnaire (in Spanish, the original language) is also provided as Appendix A.

**Figure 2 ijerph-19-07034-f002:**
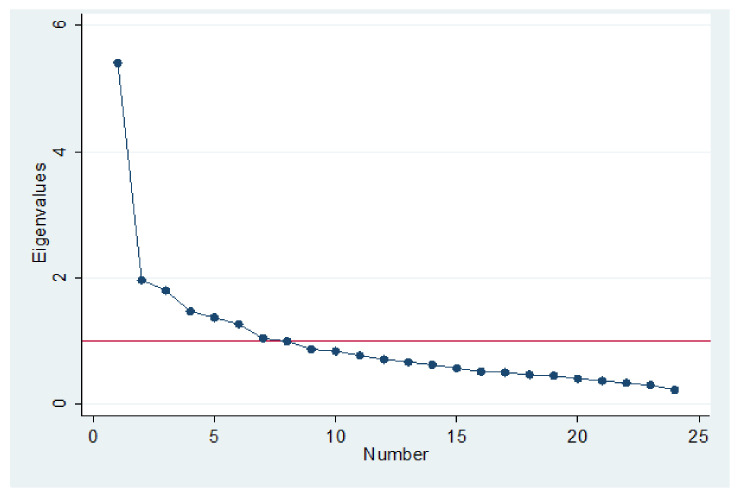
Sedimentation graph of eigenvalues after factor analysis.

**Table 1 ijerph-19-07034-t001:** Descriptive statistics of the VBDH questionnaire respondents.

	N	%
**Level of Self-Declared Knowledge about VBDH**		
Very high	4	2.74
High	33	22.60
Intermediate	71	48.63
Low	30	20.55
Very low	8	5.48
**Gender**		
Men	84	57.53
Women	62	42.47
**Age (range)**		
15–24	2	1.37
25–44	17	11.64
45–64	121	82.88
64–75	5	3.42
≥90	1	0.68
**Autonomous community**		
Andalucía	27	18.49
Aragón	2	1.37
Asturias	1	0.68
Baleares	3	2.05
Canarias	2	1.37
Cantabria	3	2.05
Castilla la Mancha	5	3.42
Castilla y León	4	2.74
Cataluña	16	10.96
Comunidad Valenciana	9	6.16
Extremadura	1	0.68
Galicia	2	1.37
La Rioja	1	0.68
Madrid	16	10.96
Murcia	5	3.42
Navarra	4	2.74
País Vasco	12	8.22
Unknown	33	22.60
**Total answers**	**146**	**100**

**Table 2 ijerph-19-07034-t002:** Cronbach’s alpha of the VBDH questionnaire items.

Item Description (*)	Cronbach’s Alpha
1. The directors of this organization see favorably to carry out changes in any area of the same to adapt and stay at the forefront of the sector.	0.891
2. The leadership in my organization favors the development of VBDH.	0.889
3. The strategies defined in the organization are aligned with the progress of VBDH.	0.889
4. The patients have participated in some way in the development of the strategy.	0.892
5. Workers are aware of the strategies and objectives of the organization in relation to VBDH.	0.888
6. Patients are aware of the objectives and initiatives that the organization carries out in relation to VBDH.	0.890
7. The organizational units are structured predominantly based on the comprehensive needs of care for the person.	0.888
8. There are alliances with companies and other public and private organizations for the incorporation of technologies that add value in the care of people.	0.887
9. The learning capacity of the workers is considered a key factor in the organization.	0.890
10. The governance model fosters a culture of trust and collaboration, promoting among professionals the value of interdependence in achieving health outcomes.	0.886
11. The voice of community agents (patient associations, business fabric, institutions, etc.) is integrated into the innovation and improvement processes.	0.890
12. Emerging technologies and innovations are identified and analyzed.	0.888
13. An ingrained part of the culture of this organization is that workers can express their opinions and make suggestions in relation to organizational and digital innovation.	0.887
14. Opportunities are systematically sought for the incorporation of digital innovation aimed at adding value to the organization.	0.887
15. All the parts that make up the organization are managed in an interconnected way, working together in a coordinated manner.	0.888
16. The information systems allow the exploitation and analysis of clinical and epidemiological data in an appropriate way for decision making.	0.890
17. There are information systems that allow the cost of the care cycle per patient to be measured.	0.888
18. There is an integrated and interoperable Electronic Medical Record for all levels of care.	0.891
19. There are initiatives for the systematic measurement of results reported by patients in clinical practice.	0.887
20. There are initiatives that allow patients to incorporate information about their health generated outside the health organization.	0.888
21. There are initiatives to systematically measure patient experience.	0.888
22. There are experiences in non-face-to-face care models.	0.890
23. There are experiences of applying Big Data aimed at improving health outcomes.	0.890
24. This organization has developed experiences to identify the patient journey with the participation of patients.	0.887
25. There are cybersecurity initiatives.	0.889
26. There are professionals trained in digital skills.	0.888
27. The organization’s professionals know the fundamentals of value-based healthcare.	0.889
28. The organization promotes experimentation and innovation on issues related to VBDH.	0.887
29. All the parts that make up the organization (departments, sections, units, work teams and individuals) are aware of how they contribute to achieving the general objectives and receive feedback on it.	0.889
30. The organization follows, and is aware of, what other organizations and entities in the sector (health, socio-health, patient associations, etc.) are doing in terms of VBDH.	0.887
31. Experiences and ideas provided by external sources (consultants, training providers, patient groups, etc.) are used as a useful tool to advance VBDH within the organization.	0.887
32. Participation in health outcomes benchmarking initiatives.	0.888
33. There is a comprehensive control panel with indicators for monitoring and evaluating costs and health outcomes per patient.	0.889
34. Periodic evaluations of data quality are carried out.	0.888
35. Access to and use of relevant information for value-based health care is timely and adapted to the needs of professionals.	0.889
36. In my organization, innovative ideas that work are rewarded.	0.887
37. The budgeted resources are sufficient for the advancement in the matter of VBDH.	0.887
38. There are economic incentive systems for professionals based on the results obtained.	0.888
39. The financing that the organization receives is aligned with the results that it obtains.	0.888
40. Payment experiences have been carried out based on health outcomes.	0.887
41. Innovative public procurement experiences have been carried out.	0.889
**Overall Cronbach’s Alpha value**	**0.891**

(*) The translation of the questionnaire items to the English language has been done only for the purposes of this publication. The original language of the questionnaire is Spanish, and this translation has not been validated.

**Table 3 ijerph-19-07034-t003:** Factor analysis results.

Factors	Items	Factor Load	Theme	Cronbach’s Alpha
1	37. The budgeted resources are sufficient for the advancement in the matter of VBDH.	0.476	Resources, incentives, and financing.	0.746
	38. There are economic incentive systems for professionals based on the results obtained.	0.780		
	39. The financing that the organization receives is aligned with the results that it obtains.	0.450		
	40. Payment experiences have been carried out based on health outcomes.	0.731		
2	4. The patients have participated in some way in the development of the strategy.	0.571	Knowledge and participation of patients and workers in the strategy of progress towards VBDH	0.632
	5. Workers are aware of the strategies and objectives of the organization in relation to VBDH.	0.636		
	6. Patients are aware of the objectives and initiatives that the organization carries out in relation to VBDH.	0.514		
3	26. There are professionals trained in digital skills.	0.361	Training of professionals and knowledge tools for advancement in VBDH	0.619
	30. The organization follows, and is aware of, what other organizations and entities in the sector (health, socio-health, patient associations, etc.) are doing in terms of VBDH.	0.547		
	31. Experiences and ideas provided by external sources (consultants, training providers, patient groups, etc.) are used as a useful tool to advance VBDH within the organization.	0.658		
4	8. There are alliances with companies and other public and private organizations for the incorporation of technologies that add value in the care of people.	0.545	Innovation initiatives	0.666
	12. Emerging technologies and innovations are identified and analyzed.	0.562		
	28. The organization promotes experimentation and innovation on issues related to VBDH.	0.384		
	36. In my organization, innovative ideas that work are rewarded.	0.447		
5	17. There are information systems that allow the cost of the care cycle per patient to be measured.	0.598	Information and quality of information	0.623
	33. There is a comprehensive control panel with indicators for monitoring and evaluating costs and health outcomes per patient.	0.474		
	34. Periodic evaluations of data quality are carried out.	0.511		
6	2. The leadership in my organization favors the development of VBDH.	0.516	Leadership, strategy, and governance	0.646
	3. The strategies defined in the organization are aligned with the progress of VBDH.	0.448		
	9. The learning capacity of the workers is considered a key factor in the organization.	0.572		
	10. The governance model fosters a culture of trust and collaboration, promoting among professionals the value of interdependence in achieving health outcomes.	0.468		
7	27. The organization’s professionals know the fundamentals of value-based healthcare.	0.492	Knowledge of the fundamentals and objectives, and access to relevant information for the VBDH.	0.570
	29. All the parts that make up the organization (departments, sections, units, work teams, and individuals) are aware of how they contribute to achieving the general objectives and receive feedback on it.	0.552		
	35. Access to and use of relevant information for value-based health care is timely and adapted to the needs of professionals.	0.414		

## Data Availability

Data are property of the Asociación Salud Digital and could be made available upon request.

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
