# Peer review of "Validation Study of a Value-Based Digital Health Questionnaire"

_ijerph, 2022, doi:10.3390/ijerph19127034_

Round 1
Reviewer 1 Report
ID: ijerph-1730570
Title: Validation study of a Value-Based Digital Health questionnaire.
Thank you for providing a chance to review this manuscript.
Detailed information:
Introduction
Line 80-85, page 2: “Taking into account the above, a questionnaire......” This sentence is complicated and confusing. Did you check the grammar? I don't know which aspects of "organizational maturity" are measured in the questionnaire in this study. Please make it clearer.
Line 80-98, page 2-3: Had there been a questionnaire or scale of this type? What aspects of organizational satisfaction did they focus on? What's innovation of your research compare to the others? Had the whole process of questionnaire development been carried out in accordance with the COSMIN guidelines?
Line 107-111, page 3: Did you have any theoretical hypothesis about the psychometric properties of the questionnaires in this study?
Materials and Methods
2.1. The VBDH questionnaire
Line 118-125, page 3: Introducing more about your questionnaire. Readers get little information about your questionnaire from the article! How did the questionnaire assess organizational maturity? Why were inclusion and exclusion criteria for participants not stated?
2.2. Sample and data collection
Line 131-132, page 3: Was the information collected only the results of this questionnaire in this study? How did you ensure that your information collection is valid in the absence of other validated and reliable questionnaires as a reference?
Moreover, if I understand correctly, you were trying to develop a new instrument for measuring the degree of maturity. We need a sufficient sample size for developing a new tool. Does the sample size really sufficient considering that your scale contains 41 items? And why are there only two sentences in a paragraph?
Results
3.1 Descriptive results
Line 155, page 4: “More than half of the participants were men (N=84; 57.53%%).” I have only seen “57.53%” instead of “57.53%%”. Did you even checked these errors before submitting?
Line 155-159, page 4: Where is the total number of participants? Are you waiting for the readers to calculate themselves?
Line 160, table 1, page 4: What is the point of Min and Max since all the variables are not continuous?
Line 167-169, figure 1, page 5: What is the point of this figure? To show the distribution of scores? What’s more, this figure is too blurry to get valid information. If you insist to put it in, improve the resolution please.
3.2 Cronbach's Alpha: internal consistency of the entire questionnaire
Line 172-173, page 6: "All items had Cronbach's alpha values greater than 0.87, and overall value was 0.891..." The digits of numbers in the article should be uniform.
Line 177, table 2, page 5: The same question as the previous comment, please unify the digits of numbers.
3.3 Exploratory factor analysis: validity analysis of subscales within the questionnaire
Line 184-196, page 7-8: Is this paragraph a discussion or a result? I suggest putting it into the discussion part.
Line 207-208, page 8: Did you really write one single sentence for one single paragraph?
Table 3, page 8-10: The presence of this table is not formatted. Three-line table (like your table 2) is what we use for reporting results.
Discussion
Line 236-237, page 10: " Sufficient information was collected to carry out an exploratory factor analysis" Is there really having sufficient information for analysis?
Line 237-268, page 10-11: The discussion is superficial. You just simply repeated the results and did not let the readers see the significance of this new questionnaire. If this is all you want to show to the global community, then your research means nothing.
With all due respect, my job is to improve the quality of a MANUSCRIPT, not a DRAFT. I did not even finish my reading. I am feeling that you were showing absolutely no respect for your valuable data. Please do not torture reviewers with such a “MASTERPIECE”.
First, reading more articles from the TOP health quality journals, to learn the formats, expressions, and of great importance—logic, might help a lot before revising this “MANUSCRIPT”.
Second, check all the possible errors. Writing a “57.53%%” in the texts makes me feel that you are asking me to point out and justify these “carelessness”, which is not possible. I am a reviewer, not a babysitter. Also, check the unity of all your writing (e.g., numbers of digits).
Third, no uniform format for all the figures and tables, such as numbers of digits, which made them look like a patchwork on a broken ship. All the figures and tables need to express the maximum, the most important information of your research.
Last, rephrase your sentences to make your expressions clear. Some of your sentences are obscure and hard to understand. Refine some of your sentences or paragraphs into a single sentence or paragraph. One paragraph only stands for one single point. And as I mentioned in the first part, logic plays a key role in scientific writing. Making your valuable data an intriguing story to tell and a coherent article for people to read, is of great importance. As presented, the writing is not acceptable for the journal. There are problems with sentence structure, verb tense, and clause construction. Furthermore, finding a native English speaker to improve the writing can considerably improve the quality.
It is a definitely great and interesting subject if could be seen by more people. But a good recipe takes patient cooking and a respectful attitude to make an appetizing dish. After completing all the above justifications and the comments of other reviewers, you may resubmit your paper.
Thank you and my best,
Your reviewer
Author Response
PLease, see the attachment.

Reviewer 2 Report
In this study, questionnaires related to value-based digital health (VBDH) were developed and their validity was verified. The importance of VBDH is increasing and its use is gradually increasing. In this respect, this study is very timely and the importance of the study is also very high.
Although it had a relatively small sample of people (N=146), its validity is very high because it is statistically processed based on thorough item analysis. However, in order to increase the completeness of the paper, the followings are to be modified.
-In 1. Introduction, introduce more related research on VBDH in other countries.
-In Chapter 2, describe whether all 150 survey participants agreed to participate in the survey.
-Describe in detail how 41 questions are classified into 5 blocks in Chapter 2.
- Explain why more than 80% of the survey participants are 45 to 64.
-It is difficult to distinguish the five divisions of arrangement in Figure 1 by color. It is recommended to use additional tables to show them in number.
-Please write 41 questionnaires in Table 2 in English as well as Spanish.
-Please describe future research issues in more detail in 5. Conclusions.
Author Response
Please, see the attachment.

Round 2
Reviewer 1 Report
Thanks for your work!
However, I think additional samples (at least 300) are needed, half of which do exploratory and the other half confirmatory factor analysis. Because the VBDH questionnaire has as many as 41 items, but the sample size is only 146. It is difficult to support the main conclusions of the study. A mismatch between the sample size and the number of item may lead to an unstable factor structure.
Author Response
Thank you for your comment. We believe the sample size is enough for the content and construct validity of the questionnaire. I agree with you that a larger sample would allow a richer analysis. However, the purpose of the analysis was to understand if the items were understood, and not to use the 41 items to generate some numeric index or numeric subscales. One objective is identifying subscales, but we are not suggesting any method to aggregate information on those subscales. For that purpose, a representative sample would be needed. To our understanding, the factor analysis conducted shows how the factor structure remains stable. Sorry if we were not clear enough with this result. In fact, when the factors with less load were excluded for the analysis the results in terms of how items aggregate is stable compared to the way they aggregate when all 41 items were included. Only the importance of the subscales varies (the alpha of subscales), but the factors that compose the subscales remain pretty stable. In addition, when including all items, there were some factors that had only one item. When we become more conservative with the factor load, and reduce the number of factors, those are allocated to another factor, but the structure -the items’ combination- looks stable. We are submitting the full Stata log file for you to revise the results as we believe this is a proof of the stability of our results. I would not have submitted the paper for peer review if I knew the sample size was not enough for the purposes of this work. It is true that a bigger sample size might allow identifying more subscales, and also would probably allow keeping more items into the analysis. We have included a paragraph regarding this limitation in the discussion section.
In lines 340-350 we have added the following text:
“However, the factor analysis shows how the factors structure is stable. In a first stage, including all the 41 items, the factors’ aggregation remains similar to the structure that remains when a more conservative approach -excluding some items according to a low factor load criterion- was selected. Some factors that had only one item, are now allocated to another factor, but when two items belong to one factor, generally they stay together, even if allocated to a different factor. This demonstrates the stability in the factor structure. Although this could be tested with a representative sample, results suggest that no big changes in the way items aggregate would be expected. However, it is true that, with a bigger sample, more factors will appear, as less items will have to be removed due to the low factor load, probably result of the small sample size in this study.”
Hopefully this is explanatory and clarifies our analysis, given the purpose of the work, but please, let us know if you still have concerns regarding the work done.
All the best, and thank you again for your time and dedication to improve our work.